# In Situ Investigation of Strain Localization in Sintered, Porous Segmented Alumina

**DOI:** 10.3390/ma14133720

**Published:** 2021-07-02

**Authors:** Vladimir Kibitkin, Mikhail Grigoriev, Alexander Burlachenko, Andrey Solodushkin, Nickolai Savchenko, Valery Rubtsov, Sergei Tarasov

**Affiliations:** 1Institute of Strength Physics and Materials Science, 634055 Tomsk, Russia; vvk@ispms.ru (V.K.); aleksburlachenko@rambler.ru (A.B.); s.ai@ispms.ru (A.S.); savnick@ispms.ru (N.S.); rvy@ispms.ru (V.R.); 2Laboratory of Nanotechnologies of Metallurgy, National Research Tomsk State University, 634050 Tomsk, Russia; mvgrigoriev@yandex.ru; 3Division for Materials Science, National Research Tomsk Polytechnic University, 634030 Tomsk, Russia

**Keywords:** segmented ceramics, pore ceramics, digital image correlation, deformation, tensor components, fracture

## Abstract

Evaporation of paraffin and ultra-high-molecular-weight polyethylene admixed with alumina powder for the slip casting and sintering process allowed the obtainment of segmented porous alumina ceramics with 50% total porosity, whose deformation behavior we studied. Structurally, these ceramic materials were composed of large and small pores, and a system of discontinuities subdividing the samples into segments. Using digital image correlation (DIC), strain distribution maps were obtained that allowed the observation of strain localization zones, where primary cracks propagated along the interblock discontinuities. Two stages were revealed to be responsible for different mechanisms that provided the sample with damage tolerance under compression loading: the first stage was crack propagation along the block boundaries, which was followed by the second stage of microcracking and fragmentation, consisting of filling of the free spaces with fragments, compaction band generation, and stabilization of the crack. Both stages comprise a cycle that is repeated again and again until the full volume of the sample is occupied by the compaction bands.

## 1. Introduction

Porous ceramics are widely used in the production of such applications as diesel particular filters, water treatment equipment, catalysts, solid oxide fuel cells, autoclaved concrete, gypsum board, bone substitute materials, and coatings—i.e., applications that require pores of different sizes and numbers [1,2,3]—and, in fact, porosity is always a balance against the material’s mechanical strength. Porous-structured brittle materials tend to reveal more complex behavior under loading compared to that of monolithic materials, where instant macroscopic fracture is a common outcome. Their fracture stage may be characterized by positive and negative dilatancy effects occurring due to the generation of voids, decompaction, and shear strain compaction [4]. These effects are responsible for the occurrence of a relatively steady pre-fracture inelastic deformation behavior in these materials. The onset of loading may be characterized either by a positive dilatancy effect during microcracking, or a negative dilatancy related to pore collapse in porous structures [4]. Another specific feature of porous ceramic materials is their deformation behavior during compression loading when the stress–strain curve deviates from a linear dependence, in a manner similar to that inherent to the plastic flow in metallic materials [3,5].

Each pore can be a stress concentrator suitable for crack nucleation and, therefore, the porosity level is a factor contributing to the behavior of ceramics under loading. In ceramics with medium isolated porosity, the onset of steady crack growth, once started from a single pore, can be changed for an unsteady fracture stage when cracks nucleated on the neighboring pores begin to merge [6]. The high-porosity ceramics with interconnected porosity demonstrated lower microcracking stress limits, and their behavior under compression is characterized by the generation of crack-like structures initiated from a group of pores with microcracks [6]. Therefore, these high-porosity ceramics are used in working conditions requiring some resistance to mechanical loading [2,7,8,9] and damage tolerance [10], so that their strength is not fully lost even upon reaching the ultimate compressive stress (UCS) and nucleation of a macrocrack. It is for this reason that these ceramics are often used in studying the inelastic “post-fracture” behavior when the material retains some number of unbroken struts, which allow it to maintain its integrity [7,8,9].

Progress has been achieved in recent years with respect to improving the damage tolerance of brittle, porous media, due to the use of numerous crack-retardation mechanisms, including that provided by microcracking [6]. Microcracking is usually accompanied by the evolution of a porous structural component when damage accumulates and finally breaks the inter-pore walls, with the ensuing filling of the empty spaces with fragments and compaction. Such a mechanism is responsible for inelastic flow in porous rocks under compression loading. This type of behavior was also observed in indenting a porous alumina, when structural breakdown resulted in pore collapse, fragmentation of grains, and the formation of compaction bands [10], following a so-called “cataclastic flow” mechanism [11]. 

It is common to discuss damage tolerance in terms of the transition from brittle to quasi-ductile behavior [12]. Deviation from the linear stress–strain behavior of brittle, porous media during the compressive loading/unloading cycle is accompanied by the appearance of a hysteresis loop. Both nonlinear behavior and hysteresis are observed due to the dissipation of mechanical loading energy by the generation of numerous microcracks instead of one main one. This microcracking mechanism allows us to observe residual strain after unloading [13]. 

It is known [14,15,16] that segmented ceramic materials can be more tolerant to defects and discontinuities generated under loading, compared to monolithic materials. Structural components of such a segmented ceramic may be either bonded together using a special binder or have a special shape that allows their pure mechanical interlocking [14,15,16]. In both cases, these segment boundaries can then serve as barriers against trans-segment crack growth and, thus, allow the segmented ceramics to maintain the post-UCS strain at a constant level—and higher than that of monolithic ceramics, due to local displacement and rotations of the constraint segments with respect to one another [14,15,16]. In fact, the segmented ceramics may be represented as being constructed of dense blocks with block boundaries, which serve for the dissipation of mechanical energy by means of microcracking, granulation, and friction.

The problem here may be that despite the basic mechanical characteristics of segmented porous ceramic materials having already been determined [14,15,16], their microstructural evolution under deformation is still yet to be analyzed, which may have an effect on the functionality of these materials. There are several methods that would allow for the study of the above-described effects in porous media. DIC [17,18] is a convenient method to visualize strain localization in porous materials [19,20,21,22,23]. In particular, DIC studies on aluminum-based foam deformation allowed the observation of a weak zone—i.e., a zone responsible for collapse—and, correspondingly, the maximum level of loading [19]. The pore wall displacement, as well as strain localization, was quantitatively analyzed with the use of DIC in porous, ultra-high-molecular-weight polyethylene (UHMWPE) [20]. 

Microstructural and mechanical changes in porous alumina ceramics under loading were investigated using both DIC and X-ray computer tomography [21]. The results were used for the optimization of structural parameters and improvement of defect distribution under dynamic loading.

Preliminary DIC experiments on the compression of porous segmented alumina showed the applicability of the method for strain localization [22,23] and, thus, paved the way for more advanced studies with the aim of observing the segmented ceramics’ evolution under loading. 

The objective of this work was to obtain new data, identifying the main features of structural evolution and strain localization in porous segmented sintered alumina under compression with the use of improved DIC visualization. 

## 2. Materials and Methods

### 2.1. Sample Preparation and Examination

A commercial alumina slip composed of 15 wt.% paraffin and 85 wt.% α-Al_2_O_3_ of mean particle size 5 μm was heated above the paraffin melting point ~70–80 °C, and mechanically stirred with 20 vol.% of 100-μm spherical UHMWPE particles until a homogeneous mixture was obtained, which was then used for injection molding and obtaining ∅10-mm- and 7.4-mm-height samples.

Preliminary thermogravimetric analysis showed that thermo-oxidative decomposition of paraffin occurred in the temperature interval 237–276 °C. When using an α-Al_2_O_3_
*+* paraffin + UHMWPE homogenized mixture, this interval was extended to that of 150–600 °C. These data were then used to carry out a stepped annealing of the green samples, as follows: The paraffin evaporation was carried out during heating to 345 °C at a rate of 15 °C/h. The UHMWPE was melted and then evaporated during the next stage by heating the samples to 600 °C at a rate of 50 °C/h. Oxygen contained in the alumina powder filler oxidized the vaporized paraffin and polymer to carbon dioxide, which then served to compact the filler. 

Thermo-oxidative decomposition of the pore-forming components allowed us to obtain a volume of gas products, which was sufficient for forming spherical pores as well as a 3D network of discontinuities, which appeared as gas-removal channels. At this stage, the sample consisted of blocks or segments. The annealing stage consisted of heating to 1000 °C at a rate of 70 °C/h, followed by maintaining that temperature for 30 min. Fully annealed green samples contained no traces of organic compounds. Sintering was carried out during heating to 1300 °C for 3 h and maintained at that temperature for another 1 h. The final 50% porosity α-Al_2_O_3_ sample was composed of recrystallized 6–10 μm grains with fine intergrain and large spherical pores, as well as a network of crack-like discontinuities, which was not completely healed during sintering. 

The large 110-μm pores were formed via the evaporation of the UHMWPE particles (Figure 1a,b). The discontinuity network subdivided the sample’s volume into ~220-μm crack-free segments (Figure 1a,b), composed of alumina grains and fine 3-μm pores.

Some of the discontinuities look to be partially bonded by bridges or struts formed by partial sintering after the evaporation of the UHMWPE. These discontinuities might have resulted from crack opening in a green sample under gas pressure from the vaporization of paraffin, which started at temperatures as low as 150 °C. Upon the full removal of both paraffin and UHMWPE, the sintering temperature of 1300 °C was achieved, which proved to not be high enough to heal the cracks formed. Then, samples were subjected to mechanical compression tests on a Devotrans GP 30KN-DLC + CKS (Turkey) universal testing machine at a loading speed of 2 × 10^−4^ s^−1^. No lubrication was applied to reduce the friction between the specimen ends and machine’s platens, except for aluminum foils, which served to protect the specimen ends against fracture. Moreover, the specimen height/diameter ratio was about 1:1 and, therefore, a triaxial stress–strain state was established throughout the specimen volume, and determined stress and strain localization.

To carry out DIC, a cylindrical surface was ground flat and polished to obtain a rectangular 7.45 mm × 7.40 mm plane field of interest (FOI). To detect possible micro- and macrodamage at different stages of deformation, samples with ground and then polished end-faces were produced. During the compression deformation, optical macroimages of the plane were taken every three seconds using a Nikon D90 camera (Nikon Corp., Japan) equipped with a macro lens, and then saved on a hard drive. 

The microstructure of the samples after sintering and compression tests was studied using a TESCAN VEGA 3 SBU scanning electron microscope (SEM) (Tescan, Czech Republic).

### 2.2. DIC Procedure

A series of FOI images was taken during the compression loading of the porous segmented ceramics, reaching a maximum stress level equal to 80% of the ultimate compression stress. The images taken at the moments of time $t_{1}$ and $t_{2}$ ($t_{2}>t_{1}$) are the reference and current images, respectively. The sample accumulated some strain during the time $\Delta t=t_{2}-t_{1}$ of compression loading, such that each small area (subset) of the current image displaced with respect to that of the reference image (template subset). Therefore, only one displacement vector value exists for each subset. Let us construct a virtual square mesh over the FOI with a space period *T* (pitch), with squares of dimensions *m* × *m* and *R* × *R (R > m)* in the current and reference images, respectively, so that these small *m* × *m* and large *R* × *R* images will be subsets and regions of search, respectively. Finding the displacement value is identical to finding a global extremum of some functional $S=\sum_{\Omega }f(I_{1},I_{2})$, where *I*_1_ and *I*_2_ are the grey scale image intensity arrays, $f(I_{1},I_{2})$ is some measure, and Ω is the local search *R* × *R* region. The result of computing the functional may be either a number or a function obtained from two arrays (or functions) according to some algorithm or equation. The measure $f(I_{1},I_{2})$ is used as a brightness difference or its correlation coefficient. In the vector head, the difference and correlation functionals must take on their minimum and maximum values on a pixel mesh, respectively. To improve the accuracy of finding the displacement vector head coordinates, a difference functional was approximated using a 3D-fitting procedure, which provided the absolute error at the level of 0.03–0.05 pixels.

The next step of the procedure is scanning over the search region of the current image with a corresponding region of the reference image. As soon as there is only one displacement vector for each subset, the functional is also capable of taking on only one value. If this is not the case, then it may mean that some inherent error occurred. The sources of errors during DIC can be identified as insufficient quality of optical patterns, high optical aberrations, sensor and circuit noise, or bad choice of DIC computation parameters. 

Normally, the FOI surface has to be as plain as possible in order to provide good focusing of the camera lens, and possess low reflectance—such that it is even common approach to paint it. The ceramic sample surfaces used in this work were not painted, but several black marks were placed for better focusing, and the input data were treated using computational parameters that provided acceptable accuracy. 

The functional value corresponding to each current position was then computed in order to determine the further coordinates of its global extremum which, in turn, determine the coordinates of the unknown displacement vector field. The program output data are two arrays of the displacement vector projections *u_x_*(*x*,*y*) and *u_y_*(*x*,*y*) on the abscissa and ordinate axes, respectively. A total vector population gives us a full displacement vector field $\vec{u}(x,y)=u_{x}(x,y){\vec{e}}_{x}+u_{y}(x,y){\vec{e}}_{y}$, where ${\vec{e}}_{x}$ and ${\vec{e}}_{y}\;$ are the relevant unit vectors.

Let us assume that the displacement vector field computed from the images obtained in moments of time *t*_1_ and *t*_2_ corresponds to the current time *t = t_2_*, while the material deformation act occurred during $\Delta t=t_{2}-t_{1}$.

For convenience, let us also consider the plastic flow at some high-vector field space period (pitch) *T,* so that *T* will be a number of pixel-size intervals between two vectors; normally this period is selected from an inequality, as follows: 10 ≤ *T* ≤ 100. To reveal more details of the plastic flow pattern, it is reasonable to assume the *T* value within a 1≤T≤3 interval, thus balancing the refinement against the computation time, which is proportional to *T^2^*. 

Strain tensor components were computed according to solid-state deformation equations, as follows:$$\varepsilon_{xx}=\partial u_{x}/\partial x,\;\varepsilon_{yy}=\partial u_{y}/\partial y,\;\varepsilon_{xy}=(\partial u_{x}/\partial y+\partial u_{y}/\partial x)/2.$$

Then, principal shear strain γ(x,y) was determined at the point of interest according to Equation (1):(1)$$\gamma =\sqrt{{\left(\varepsilon_{xx}-\varepsilon_{yy}\right)}^{2}+4\varepsilon_{xy}^{2}}.$$

Spatial distribution of strain values can be visualized as so-called “pseudo-images”, using a procedure as follows: Let the strain change in the range $\gamma_{\min}^{}\leq \gamma (x,y)\leq \gamma_{\max}$, where $\gamma_{\min}^{}$ and $\gamma_{\max}$ are the minimal and maximal strain computed from the displacement vector field, respectively. The 8-bit image brightness values lie inside the 0≤I(x,y)≤255 interval, and nothing prevents us from applying the same gradation for the strain interval and then matching the strain to the corresponding grey scale intensity at each point of the image. If the high strain is matched to low brightness at a point, the resulting mapped image will then be an inverse pseudo-image. This procedure of mapping the strain distribution into strain distribution maps (pseudo-images) can be represented as follows: (1) calculate strain at each point of the displacement field according to Equation (1); (2) if the calculated strain is zero, then assign it a value one order of magnitude lower than that of minimal non-zero $\gamma_{\min}^{}$; (3) take a logarithm of the strain values; (4) subdivide the total range of values into 255 intervals; (5) substitute the strain value for the corresponding minimal interval number; and (6) reconstruct the pseudo-image of strain distribution from the corresponding gray gradation levels. The above-described procedure allows us to obtain more informative and intuitive visualization of the strain distribution over the FOI.

Estimation of the microcracking mechanism was carried out by computing and analyzing rate-of-strain tensor (ROST) components according to the following procedure: Displacement rate values were determined using the equation $v_{y}(y,\varepsilon )=u_{y}(y,\varepsilon ,x=X_{m}/2)/\Delta t$, where ε=Δh/h is the total strain, $X_{m}$ and $Y_{m}$ are the specimen’s width and height, and Δ*h* is the specimen elongation (Figure 2). To determine the ROST components ${\dot{\varepsilon }}_{xx}$ and ${\dot{\varepsilon }}_{yy}$, a procedure was applied consisting of several steps, as follows: (1) constructing dependences $v_{x}(x,\varepsilon )=u_{x}(x,\varepsilon ,y=Y_{m}/2)/\Delta t$ and $v_{y}(y,\varepsilon )=u_{y}(y,\varepsilon ,x=X_{m}/2)/\Delta t$; (2) smoothing each of them by adjacent averaging; (3) coordinate differentiation; and (4) smoothing again. 

Finally, the ROST components were obtained, as follows: (2)$${\dot{\varepsilon }}_{y}(y)=\frac{\partial v_{y}(y,\varepsilon =const,x=X_{m}/2)}{\partial y},\;{\dot{\varepsilon }}_{x}(x)=\frac{\partial v_{x}(x,\varepsilon =const,y=Y_{m}/2)}{\partial x}.$$

## 3. Results

### 3.1. Compression

The response of a sample to both monotonic (Figure 3a) and cyclic loading (Figure 3b) can be used for qualitative estimation of its elasticity. The macrographs taken from the cylindrical surface of the sample under compression (Figure 3a) showed that the first cracks appeared on reaching a stress level corresponding to 0.3 fraction of the ultimate compression strength (UCS) and, therefore, this material could be characterized as having low damage tolerance. 

In addition, there is a “post-fracture stage” on reaching the UCS. In other words, the stress–strain curve allows us to observe some inelastic behavior of the porous ceramics, which is different from that of traditional brittle materials, with catastrophic fracture (disintegration) after the nucleation of the very first crack.

A total of six loading/unloading cycles achieving 0.8 UCS stresses were taken into consideration (Figure 3b). A hysteresis with notable residual strain could be observed after the first loading/unloading cycle. The following loading/unloading cycles did not result in residual strains. In addition, increasing the number of cycles resulted in an increase in the tilt angle of the curves with respect to the abscissa axis, which is evidence of an increase in the elasticity modulus value. 

The compressive UCS of the porous segmented alumina was 34 MPa and, according to results reported elsewhere [7], this value may be considered satisfactory for 50% porous samples, despite their segmented structure and network of discontinuities partially healed by low-temperature sintering. The DIC studies were therefore focused on the ascending branch of the stress–strain diagram, from 0.1 to 0.8 UCS stress levels (Figure 3c,d). The high peak in the dσ–dε plot located within the 0–0.01 strain range (Figure 3d) can be also explained by accommodation between the sample and the compression machine platens, with aluminum foil placed between them in order to prevent the edge from fracturing.

### 3.2. DIC

#### 3.2.1. In Situ Strain Mapping and SEM

It takes rather a lot of computation time to obtain a displacement vector field for pitch *T* = 1, but this is the price to be paid for constructing detailed, high-resolution strain distribution maps (SMs), which are very convenient for analyzing the evolution of the porous ceramics under loading (Figure 4). 

Inelastic strain localization begins with the generation of a vertical primary macroband (Figure 4a–c), while no crack is yet observed in the corresponding optical images (Figure 4a’–c’). Further loading results in the generation of complementary strain localization macrobands (SLMBs) (Figure 4d–f), with their orientations deviating from the vertical one. Only at this stage of the loading do the optical images (Figure 4d’–f’—see inside the darker area along the vertical centerline) allow the observation of a primary crack oriented along the primary SLMB. The primary crack starts propagating from a system of small cracks formed near a central black mark on the FOI, which is related to the strain localization zone in Figure 4a.

The presence of inherent cracks offers easy routes for the development of the primary crack, so that it can easily grow by breaking the only partial bonds between the crack-free segments. On the other side, these segments can independently slide against one another under loading and generate fragments. Such a process is analogous to that occurring during external friction between two hard, brittle bodies. 

The strain visualization maps obtained from specimens deformed at higher strain levels demonstrate further generation of the complementary bands, and simultaneous attenuation of the primary ones, so that almost the entire FOI area becomes involved in deformation (Figure 5). Both primary and complementary SLMBs still can be observed only in Figure 5a,b, and then almost completely disappear in Figure 5c. The corresponding optical in situ images show neither notable opening of the primary cracks nor formation of the extra ones. As a matter of fact, the image in Figure 5c’ allows suggesting some primary crack closure, which also will be shown later using rate-of-strain tensor component distribution.

Microcracking, fragmentation, and compaction of the intersegment spaces (discontinuities) may be illustrated by the SEM BSE images in Figure 6a, where wide compaction bands formed along the discontinuity network, with fragmentation and filling of the interblock spaces with 10–20-μm fragments (Figure 6b–d). 

Two types of compaction bands formed under loading—primary compaction bands, which run across the FOI (shown by arrows in Figure 6a); and accommodation compaction bands formed around the segments (shown by arrows in Figure 6b,c). It can be seen from Figure 6b–d that the former large pores are filled with small fragments as a result of microcracking.

#### 3.2.2. Estimation of Microfracture-Retardation Mechanism by Rate-of-Strain Tensor (ROST) Components 

The evolution of the ROST components along the corresponding coordinate axes, as dependent on the total strain, is represented in Figure 7a,b. The transversal ROST component ${\dot{\varepsilon }}_{xx}(X,Y=Y_{m}/2)$ demonstrates the evolution of the crack when the crack starts to open at low strain, reaches its maximum, and then starts to close (Figure 7a). The longitudinal component ${\dot{\varepsilon }}_{yy}(Y,X=X_{m}/2)$ distribution shows how this crack’s edges experience microcracking and fragmentation (Figure 7b).

The evolution of the distances between peaks may correspond to that of the block size. The interblock spaces become less clearly observed at higher strain, and to make them more visible the following procedure was used: Four ${\dot{\varepsilon }}_{yy}$ curves were taken into consideration as corresponding to some sequential moments of time, and the distance ***a*** between two points where at least three curves intersected was then assumed to correspond to the block’s size (Figure 8a). Applying such a procedure repeatedly, it was possible to obtain a series of block size values, which then were treated statistically in order to obtain a dependence of the mean block size on the strain, as well as corresponding standard error (SE) values (Figure 8b). Mean block size was obtained at the level of ~220 μm, with an SE as high as ~100 μm. The deformation component ${\dot{\varepsilon }}_{yy}$ behavior can be interpreted as resulting from the fact that some of the cracks were opening while others were closing during loading. The negative values of this deformation tensor component were dominant, thus indicating that cracks were mainly closing, but at the same time the block fragmentation was related to the opening of cracks. It can be seen from Figure 8b that the first fracture stage was even accompanied by some increase in the mean block size—up to 260 μm—while later stages demonstrated its reduction to ~210 μm.

The oscillating mean block size’s dependence on the total strain may be explained by strain accumulation because of three mechanisms—fracture, microcracking/fragmentation, and fragment compaction with generation of ~50-μm-width accommodation bands by the block boundaries (Figure 5b,c).

Considering the evolution of the maximal and minimal values of the ROST components, it is possible to delineate some evolutionary stages (Figure 9). These stages were determined only for the purposes of discussion, because of the great scatter of the data and the triaxial stress–strain state.

The maximum ${\dot{\varepsilon }}_{xx}$ component value (Figure 9, curve 1) shows the highest peak, which may be related to the stage of the primary crack’s nucleation, growth, and closing. As long as the component values stayed positive, the crack continued to open in response to further loading. The negative minimum values (Figure 9, curve 2) of the component may be evidence that some other smaller cracks that formed away from the primary one were closing. Such behavior may be the result of an inhomogeneous triaxial stress-–strain state. The absolute values of both maximum and minimum components became close in the strain interval 0.020–0.036, with the exception of the high negative peak at ~0.0033 on the minimum value curve, which indicates the closure of some portion of the primary crack. Further evolution of both the maximum and minimum values indicated almost the same likelihood of opening and closing events. It is worth noting that despite a great scatter, the elongation component values varied within a range of 3–4 orders of magnitude, and some correlation to the fracture stages can be observed.

#### 3.2.3. Fracture-to-Compaction Banding Transition

The above-described evolution of the segmented porous ceramics under compression loading may be roughly divided into two stages: The first stage may be related to brittle fracture when the primary crack nucleates and grows in accordance with the stress localization and concentration. Such a stage is similar to that observed in brittle, homogeneous materials, and can be defined as an inhomogeneous deformation stage. The second stage may be associated with the generation of primary and accommodation compaction bands, progressive homogenization of the inelastic flow, and uniform distribution of the strain. Therefore, this stage can be characterized by some mean strain, whose evolution may be related to the above-identified inelastic deformation mechanisms.

Given that the displacement field is homogeneous, the strain γ values at each point of the FOI must also share the same value. Strain tensor components and vector field components will then be functions of their coordinates, as follows:(3)$$u_{x}^{}(x,y)=a_{x}\cdot x+b_{x}\cdot y+c_{x}\;\;\;u_{y}^{}(x,y)=a_{y}\cdot x+b_{y}\cdot y+c_{y},$$
where $a_{x},\;b_{x},\;c_{x},\;a_{y},\;b_{y},\;\mathrm{and}\;c_{y}$ are the constants to be determined from the experiment, $u_{x}^{}(x,y)$ and $u_{y}^{}(x,y)$ are the experimentally measured displacement components specified as $u_{x}^{}(i,j)$, $u_{y}^{}(i,j)$ arrays with M and N lines, respectively, and i and j are line and column numbers, respectively.

Each of the arrays can be approximated by a plane in order to obtain the above-mentioned constants, as well as the strain rate averaged by the FOI, as follows:(4)$${\dot{\gamma }}_{mean}=\sqrt{{\left({\dot{\varepsilon }}_{xx}-{\dot{\varepsilon }}_{yy}\right)}^{2}+4{\dot{\varepsilon }}_{xy}^{2}}=\sqrt{{\left(a_{x}-b_{}{{}_{y}}_{}\right)}^{2}+{\left(b_{x}+a_{y}\right)}^{2}}/\Delta t$$

Mean squared deviation ${\dot{\sigma }}_{mean}$ (MSD) was also computed to evaluate deviation from the corresponding plane, as shown below:$${\dot{\sigma }}_{mean}=\sqrt{D_{x}+D_{y}{}_{}}/M\cdot N$$
(5)$$D_{x}=\sum_{i=1}^{M}\sum_{j=1}^{N}({\left(u_{x}^{a}\right)}_{i,j}-{\left(u_{x}\right)}_{i,j}{)}^{2},\;D_{y}=\sum_{i=1}^{M}\sum_{j=1}^{N}({\left(u_{y}^{a}\right)}_{i,j}-{\left(u_{y}\right)}_{i,j}{)}^{2}$$
where $u_{x}^{a}$ and $u_{y}^{a}$ are the displacement values determined after approximation. The MSD was normalized both by time and the number of displacement field points, for more clear visualization of the deformation stages; it can therefore be understood as a degree of displacement field deviation from that of homogeneous deformation, or as a ratio between the actions of fracture and inelastic deformation mechanisms. The reliability of these data was additionally improved by averaging over the M×N=441⋅425 array.

The mean strain rate ${\dot{\gamma }}_{mean}$ and corresponding MSD peaks during evolution phase I (Figure 10) may be related to the primary crack opening and closing because of the transition to microcracking and positive dilatancy at phase II, and then to generation of the primary compaction bands at phase III. The MSD values during phase I stay high because of normal crack opening, which is also a reason behind the drop in the dσ–dε curve.

Next, phase IV is characterized by MSD peaks that appear due to secondary cracks opening in some FOI zone, which then changes again for microcracking at phase V, and compaction banding at phase VI. Phase VII consists of crack growth once more. Mean strain rate ${\dot{\gamma }}_{mean}$ behavior is similar to that of MSD, but it is less sensitive to the alternation of deformation mechanisms, as well as that of the dσ–dε curve behavior.

It can be assumed that phases IV–VI physically describe the same process as stages I–III, but are related to accommodation strain localization bands instead of primary ones. Phase VII consists of cracking again, i.e., the onset of another cycle.

## 4. Discussion

### 4.1. Compression

It is a well-known fact that the energy of elastic deformation accumulated by fully elastic material in loading is fully released during an unloading stage. The non-elastic behavior is accompanied by dissipation of this energy and deformation behavior hysteresis. For example, this energy can be spent in the generation of new surfaces in the low-temperature loading of brittle ceramic samples.

The as-sintered porous and segmented ceramics show rather wide hysteresis loops during the first loading/unloading cycle, as well as large residual strain (Figure 3b). The rationale here may be that cracks nucleate, open, and propagate via the network of crack-like discontinuities when blocks displace with respect to one another.

Under cyclic deformation conditions, some porous ceramic materials and rocks increase their Young’s modulus values, depending on the stage of the loading/unloading cycle [13,24,25]. Such behavior is commonly interpreted in terms of the crack opening/closing and the generation of compaction bands.

The preliminary characterization of porous segmented ceramics for strength and elasticity tells us about the existence of a structure with low damage tolerance and damage accumulation that allows efficient dissipation of mechanical energy by means of compaction banding. All of the above-mentioned processes may occur at the ascending part of the compression loading curve, up to reaching 0.8 UCS stress; this is exactly where the behavior of the dσ–dε curve in Figure 3d shows numerous peaks and troughs that may be related to microscale stress concentration and fracture events.

### 4.2. Primary and Accomodation Compaction Bands

The orientation of SLMBs and cracks depends on the stress–strain state formed in the material under compression loading. It was indicated above that a triaxial stress state is formed in both the top and bottom parts of the sample under compression loading because of the lack of lubrication between these surfaces and the testing machine platens. The use of a DIC procedure allowed us to obtain a two-dimensional projection of this state on the specimen face.

Another aspect is that both friction and fracture are developed on the inherent crack surfaces during the inelastic deformation of hard solids such as ceramics, rocks, or granular media [13,24,25]. The presence of stress concentrators, crack edges, interfaces, or boundaries with full or partial bonding between them offers an opportunity for the development of shear along them, which is always conjugated with friction and friction-induced evolution of the near-interface zones. More homogeneous distribution of strain at later stages of the compression loading may be related to the action of such a mechanism when strain localization on the inherent discontinuities results in sliding friction between the ceramic segments, their fragmentation, and the refining and filling of the interblock spaces with debris (Figure 6b–d). The resulting stress relaxation and increased friction caused the full arrest of the sliding in this particular zone, switching it to some other zone. Such a relay process might result in the homogenization of the strain distribution, and the effective dissipation of mechanical energy, with the generation of compaction bands.

In fact, compaction banding is an alternative to crack growth, which starts just after primary crack growth. It can be seen from Figure 10 that MSD peaks related to crack growth alternate, with relatively low MSD curve portions provided by compaction banding. Such a switching mode defines strain accumulation during inelastic deformation.

The results obtained in this work show that segmented, porous ceramics may reveal deformation behavior similar to that observed in limestone or in granulated media, when strain localization is manifested in the form of compaction or shear bands [26,27,28]. These bands may be described as long, narrow, almost plain shear strain zones accommodated by dilatation, compaction, etc. [29].

The generation of shear bands in granulated media under deformation is also a well-known phenomenon, which may be related to particle rolling and fragmentation [30]. Particle fragmentation rate has its effect on the generation of shear bands; for example, no shear band localization can occur in a material prone to easy fragmentation; instead, a distributed system of strain localization zones might appear. [31,32].

Taking into account all of the results obtained, the evolution of segmented, porous alumina with loading may be described as follows: Strain localization in the sample under loading results in crack propagation by the inherent cracks until the formation of a primary crack, whose preferential orientation is determined by a triaxial stress–strain state established in the specimen under compression loading. This primary crack is really an interface now, allowing two parts of the specimen to slide with respect to one another and cause fragmentation of the edge material, until becoming a sort of compaction band filled with fragments. Friction in such a compaction band becomes too high, and further propagation of the crack is arrested. The strain is redistributed and a new localization zone appears, which follows the same route until almost all of the specimen’s volume is occupied by arrested compaction bands, including that of the coarse pores.

Segmented, porous alumina was obtained in this work possessing a topological structure with rather weak bonding between the blocks, which allows the sample’s integrity to be retained due to compaction banding. Another advantage is that the material’s segments are capable of energy dissipation via displacing with respect to one another, microcracking, and crack retardation. This mechanism allows the combination of acceptable mechanical strength and fracture tolerance under mechanical loading, and therefore has potential in a wide range of applications—such as filter media, membranes, and bone tissue endoprostheses—where functionality must be combined with acceptable mechanical strength and processability.

### 4.3. Future Efforts

Further work may be devoted to developing a unique inner geometry and structure of the ceramics that would allow us to take advantage of an organized, hierarchical structure composed of three components, as follows: (1) a segmented structure to control crack propagation during compression loading; (2) a large pore structure to control the segment’s size; and (3) polycrystalline ceramic grains with small pores to control microcracking, filling of the pores with fragments, and compaction. All of these components can be tailored by adjusting the sintering parameters.

A promising method for building special porous and segmented structures is selective laser sintering [33,34,35,36]. Using such an approach, it would be possible to synthesize structures composed of polycrystalline porous segments and large pores, which would generate a desired pattern of compaction bands under compression loading. The large pore structures would then serve for the collection of microcracking debris and the redistribution of stress.

It was mentioned above that the fracture mechanisms observed in porous, segmented alumina are similar to those observed in the compression of porous rocks. This similarity allows the combination and transfer of knowledge between material science and mineralogy in terms of the structures and inner geometry of natural and synthesized ceramic materials. Such a geo-inspired conception was successfully applied when developing, for example, synthetic zeolites for catalysis and gas waste separation [37]. Along with that, this conception can be used for improving the mechanical characteristics of ceramics, by creating structures capable of imitating the “cataclastic” behavior seen in natural porous media.

## 5. Conclusions

The behavior of sintered, porous, segmented, 50% porosity alumina under uniaxial compression loading has been studied using the DIC method. The sintered alumina was structurally composed of polycrystalline alumina grains with interior ~3-μm pores, a network of discontinuities that subdivided the sample into ~220-μm segments. And ~110-μm pores located at the discontinuity network nodes. Both the bimodal pore structure and the discontinuity network were the results of the evaporation of paraffin and ultra-high-molecular-weight polyethylene admixed with alumina powder via slip casting. Only partial bonding bridges between the segments were formed during a low-temperature sintering at 1300 °C for 3 h.

In situ strain localization was analyzed by mapping the obtained strain distribution into 255 gray gradation images and comparing them with the corresponding SEM images.

As shown at the first stage of compression testing, the strain was localized on the discontinuities, thus leading to primary crack growth (Phase I). Phase II was characterized by microcracking and fragmentation on the segment boundaries along the crack length, with filling and compaction of the interblock spaces and large pores with fragments, thus forming a compaction band (Phase III), which highly increased friction between the crack surfaces, retarded the primary crack, and then initiated cracking in some other interblock spaces. This cycle was then repeated several times until the entire sample was occupied by the compaction bands. Such inelastic behavior by microcracking, fragmentation, and compaction banding provided efficient stress relaxation in the porous, segmented alumina, thus increasing its damage tolerance.

## Figures and Tables

**Figure 1 materials-14-03720-f001:**
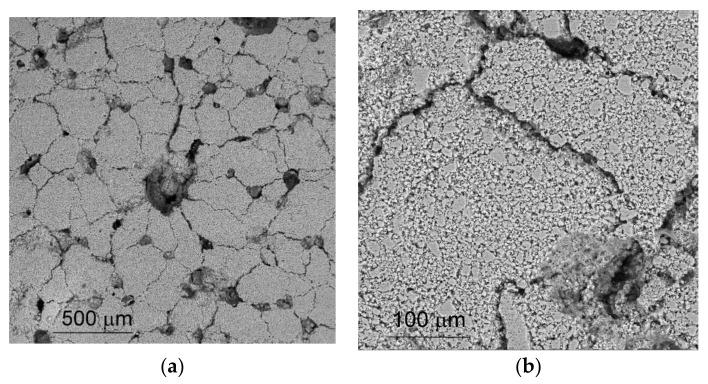
The SEM images of as-sintered porous segmented ceramics (**a**) and polycrystalline block structures (**b**).

**Figure 2 materials-14-03720-f002:**
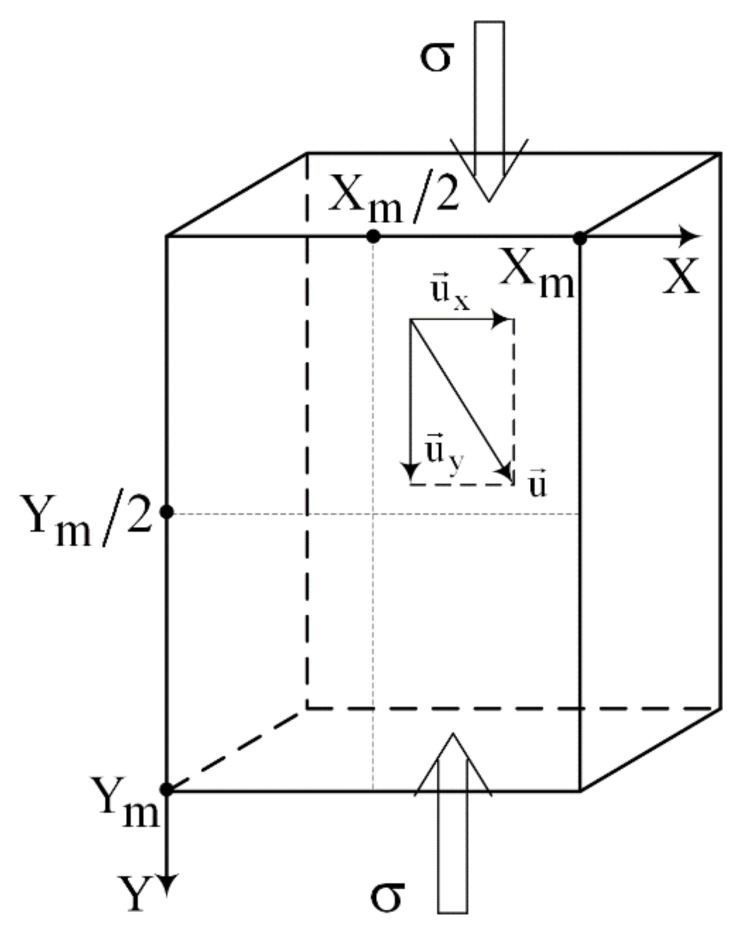
Compression loading diagram and coordinate system.

**Figure 3 materials-14-03720-f003:**
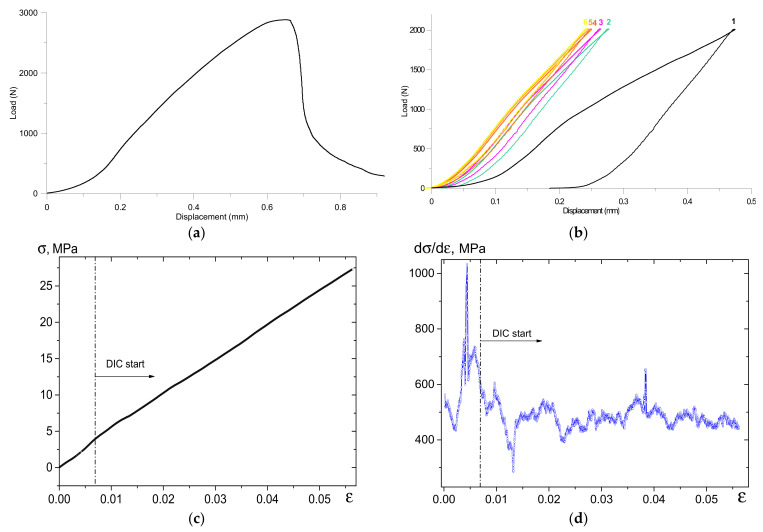
The monotonic compression load–displacement curve (**a**); cyclic loading (**b**); stress–strain diagram below the 0.8UCS (**c**); and the behavior of the dσ–dε curve below 0.8UCS (**d**).

**Figure 4 materials-14-03720-f004:**
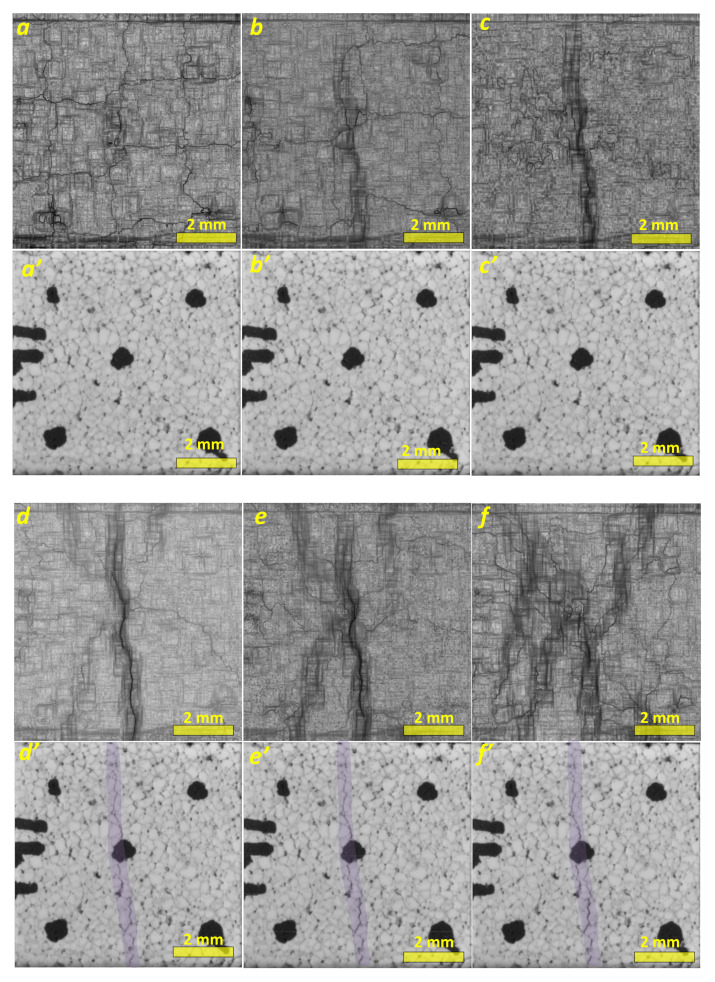
In situ strain localization maps (**a**–**f**) and corresponding optical macrographs of segmented porous alumina (**a’**–**f’**), at strain levels as follows: ε=0.0012, Δε=0.0012 (**a**); ε=0.0024, Δε=0.0024 (**b**); ε=0.0042, Δε=0.0042 (**c**); ε=0.0072, Δε=0.006 (**d**); ε=0.0102, Δε=0.006 (**e**); and ε=0.0144, Δε=0.006 (**f**). Black marks were drawn to better focus the camera.

**Figure 5 materials-14-03720-f005:**
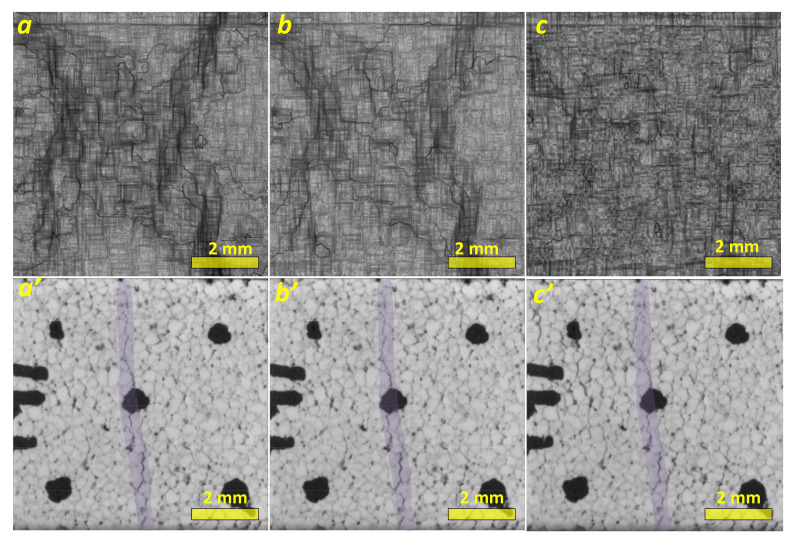
In-situ strain localization maps (**a**–**c**) and corresponding optical macrographs of segmented porous alumina (**a’**–**c’**) at strain levels as follows: ε=0.0222, Δε=0.006 (**a**); ε=0.0282, Δε=0.006 (**b**); and ε=0.045, Δε=0.006 (**c**).

**Figure 6 materials-14-03720-f006:**
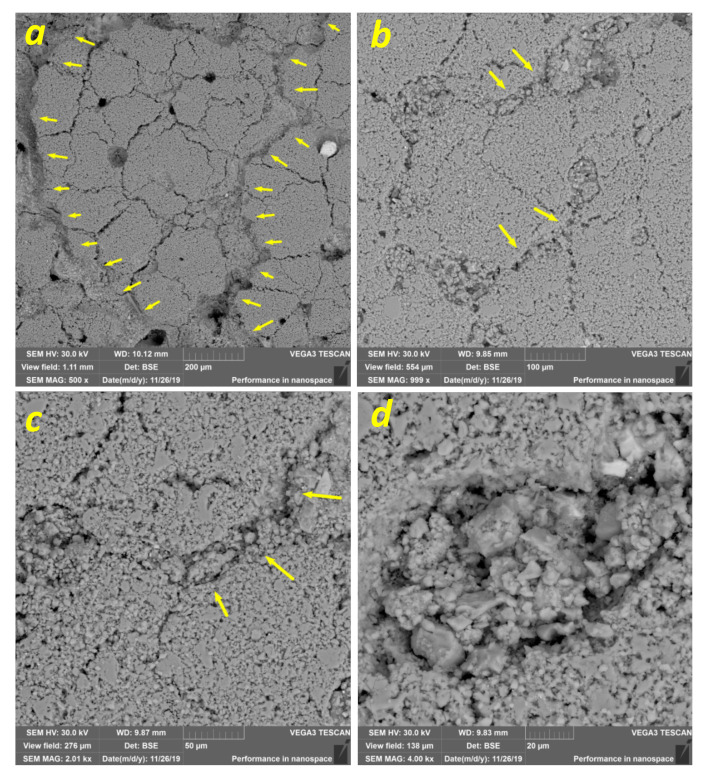
Ex situ SEM BSE images of primary compaction bands (**a**), accommodation compaction bands formed by fragment compaction in the interblock spaces (**b**,**c**), and former pores filled with small fragments as a result of microcracking (**d**) at 0.8 of ultimate compaction stress.

**Figure 7 materials-14-03720-f007:**
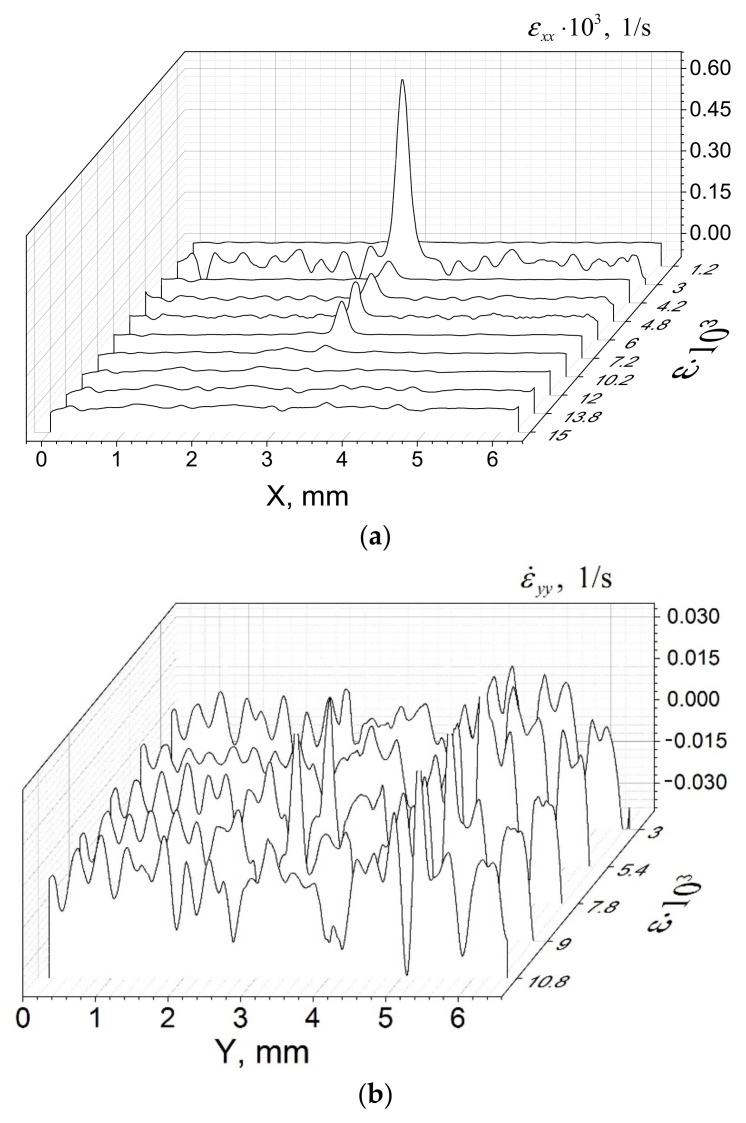
Evolution of rate-of-strain tensor (ROST) components ${\dot{\varepsilon }}_{xx}(X,Y=Y_{m}/2)$ (**a**) and ${\dot{\varepsilon }}_{yy}(Y,X=X_{m}/2)$ (**b**); Δε = 0.003.

**Figure 8 materials-14-03720-f008:**
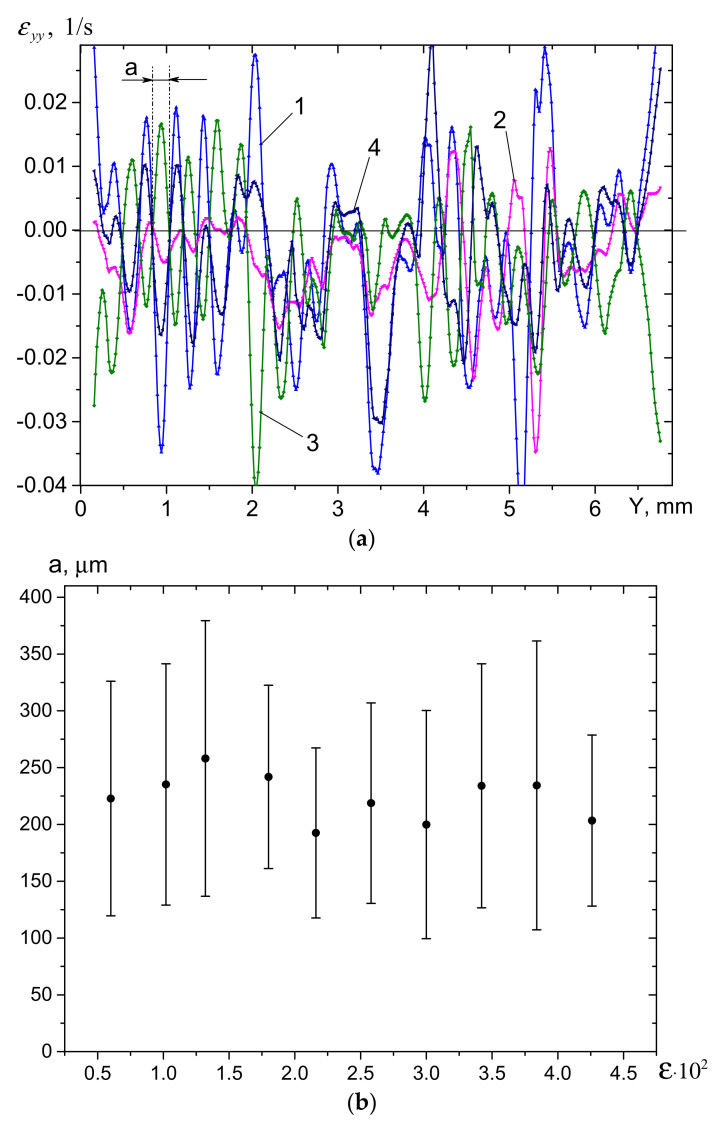
In situ distribution of ${\dot{\varepsilon }}_{yy}(Y,X=X_{m}/2)$ values (**a**) and the mean block size evolution with the strain (**b**). 1 − ε = 0.0156; 2 − ε = 0.0174; 3 − ε = 0.0186; 4 − ε = 0.0204; ∆ε = 0.003.

**Figure 9 materials-14-03720-f009:**
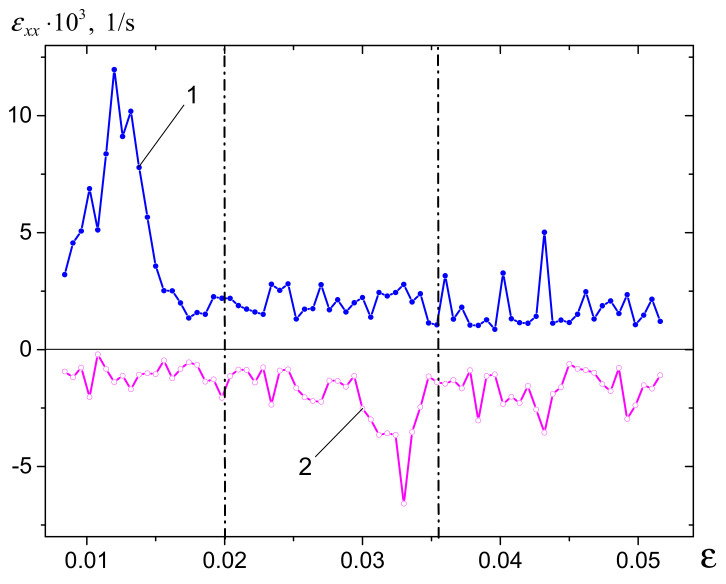
In situ evolution of maximum (1) and minimum (2) rate-of-strain tensor components ${\dot{\varepsilon }}_{xx}$.

**Figure 10 materials-14-03720-f010:**
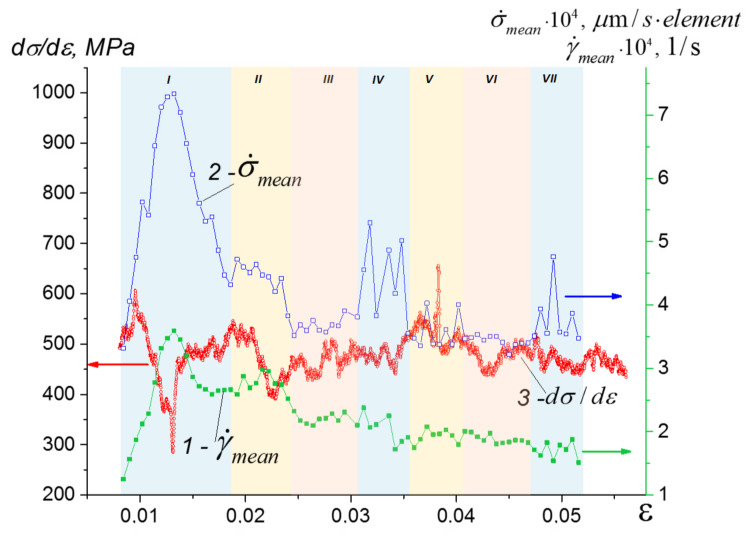
In situ evolution of the strain rate ${\dot{\gamma }}_{mean}$ (1) averaged over the vector field area, mean squared deviation for the displacement rate (2), and dσ–dε (3).

## Data Availability

Data sharing is not applicable to this article.

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
