# Peer review of "In Situ Investigation of Strain Localization in Sintered, Porous Segmented Alumina"

_materials, 2021, doi:10.3390/ma14133720_

Round 1

Reviewer 1 Report

Title: "In-situ investigation of strain localization in sintered porous segmented aluminа"

In this work the quasi-plastic behavior of sintered porous segmented aluminа under compression loading has been studied using digital image correlation (DIC) method. More specifically, the alumina ceramics contained 50 vol.% of 110 μm and 3 μm pores, as well as discontinuities segmenting the sample into 220 μm blocks or segments. In particular, the authors analyzed the strain localization by mapping the obtained strain distribution into 255 gray gradation images and comparing them with corresponding SEM images. This work shows that the strain was localized on the discontinuities during loading thus leading to primary crack growth. More specifically, quasiplastic behavior started when a compaction banding mechanism was initiated and lead to crack closing and more homogeneous strain distribution. In addition, the authors claim that the inter-block spaces were filled with the fragments generated by microcracking on the block boundaries during their displacement with respect each other. As a consequence, this sliding process highly increased friction between the crack surfaces and finally served to arrest the crack and initiate cracking in some other inter-block space. In this way, the fragmentation/compaction process occupied the total sample area. Such a quasi-plastic behavior by microcracking, fragmentation allowed efficient stress relaxation and thus increased the damage tolerance.

General comment: The main aim of this work is to obtain new data on quasi-plastic deformation and strain localization in porous segmented sintered alumina using an improved DIC visualization.

However, this work should be revised to enhance its quality and impact. Some parts are not clear and should be clarified to better understand the logic flow and the real potential of this work.

In particular, the manuscript sections should be better organized and not mixed, as in the current version of the work. More specifically, all equations, which are not results, should be inserted in the “Methods” sections, while all the results should be inserted within the “Results” section, ideally without comments. A “Discussion” section should be inserted before the “Conclusion” section. In this important section, all the results should be discussed and compared to the current state of the art. Several figures, as explained in the following section, should be reworked in order to provide high resolution images, with the correct size and with a full explanatory caption. Figures are important since they could provide a lot of information. Therefore their quality should be carefully checked. Finally, the quality of the language could be further improved.

Some specific comments:

Lines: “The samples were subjected to mechanical compression tests on a Devotrans GP

30KN-DLC+CKS (Turkey) universal testing machine at a loading speed of 2×10-4 s-1. No

lubrication was applied to reduce this friction and increase the uniaxially loaded central

part of the sample. Also specimen height/diameter ratio was about 1:1 and therefore, tri

axial stress-strain state was established throughout the specimen volume and determined

stress and strain localization.

To carry out DIC, a cylindrical surface was ground flat and polished to obtain a rec

tangular 7.45 mm × 7.4 mm plane field of interest (FOI). To detect possible micro- and

macro-damages at different stages of deformation, samples with ground and then pol

ished end-faces were produced. During the compression deformation, optical macro-im

ages of the plane were taken every three seconds using a Nikon D90 camera (Nikon Corp.,

Japan) equipped with a macro lens, and then saved on a hard drive.

The microstructure of the samples after sintering and compression tests was studied

using a scanning electron microscopy SEM TESCAN VEGA 3 SBU (Tescan, Czech Repub

lic).”

*) Probably a scheme of the experimental set-up could be beneficial to interested readers

Lines: “2.2. DIC procedure”

An image of the FOI that has been obtained and saved on a hard drive before any

compression loading is called here an image of reference. On mechanical loading another

image is obtained from the same FOI that is called a current image. The image of reference

and current image were then segmented into a mm subset and related RR (R>m) search

regions of virtual mesh, respectively. The existence of a global extremum of the functional

is evidence of finding the vector head coordinates, where I1 and I2 are the

grey scale image intensity arrays; f(I1,I2) is some measure; and Ω is the size of a local search

region. Next stage is related to scanning by corresponding region of the reference image

over the search region of current image. The functional value corresponding to each current position is then computed in order to determine further coordinates of its global extremum which in their turn, determine the coordinates of the unknown displacement vector head. A sub-pixel accuracy of the DIC results was achieved using a bicubic interpolation scheme for the S functional. The program output data are two arrays of the displacement vector projections ux(x,y) and uy(x,y) on the abscissa and ordinate axes, respectively.

A total vector population gives us a full displacement vector field

where and are the corresponding unit vectors.

Let us assume that the displacement vector field computed from the images obtained

in moments of time и corresponds to the current time , while the material

deformation act occurred during .

For convenience let us also consider the plastic flow at some high vector field space

period (pitch) T, so that T will be a number of pixel-size intervals between two vectors,

normally this period is selected from the inequality as follows . To reveal

more details of the plastic flow pattern it is reasonable to select it inside an interval

, thus balancing the refinement against the computation time, which is proportional to .

Strain tensor components were computed according to the well-known solid state

deformation equations as follows:

*) The procedure detailed here by the authors is not clear. Please improve the description of the procedure for interested readers. In particular, it is not clear why “The existence of a global extremum of the functional is evidence of finding the vector head coordinates, where I1 and I2 are the grey scale image intensity arrays; f(I1,I2) is some measure; and Ω is the size of a local search

region. “ What is “ the vector head coordinates” ? What form has the claimed functional ? What are the conditions under which the global maximum exists ? Why not several local maxima and just one global maximum. Please provide a clear and complete explanation.

In addition, some further details should be provided for the lines:”A sub-pixel accuracy of the DIC results was achieved using a bicubic interpolation scheme for the S functional. The program output data are two arrays of the displacement vector projections ux(x,y) and uy(x,y) on the abscissa and ordinate axes, respectively. A total vector population gives us a full displacement vector field

where and are the corresponding unit vectors. Let us assume that the displacement vector field computed from the images obtained in moments of time и corresponds to the current time , while the material deformation act occurred during .

Lines:” Figure 2. The monotonic load/displacement curve (a); cyclic loading (b); stress/strain diagram below the 0.8UCS (c); the

behavior of the dσ/dε curve below 0.8UCS stress (d);

The Figure 2b is not clear. Indeed, what are the initial displacement of the 2th, 3th, etc.. cycles ? In general a cycle starts from displacement=0 and the end is ruled by the amount of the plastic deformation of the material. In this case the first cycle seems to end at 0.2 mm, while the 2-th seems to start from 0.3 mm. The same for the next ones. Please explain this phenomenon in a better way. Similarly, Figs. 2C and 2d are not clearly explained.

Paragraph 3.2. DIC, 3.2.1. Strain mapping

Lines: “It takes rather much computation time to obtain a displacement vector field for pitch

T=1 but it is price to be paid for constructing a high-resolution detailed strain distribution

maps (SM), which are very convenient for analyzing evolution of the porous ceramics

under loading (Fig. 3).

Quasi-plastic strain localization starts from generating a vertical primary macroband

(Fig.3a, b, c) while no crack is observed yet in corresponding optical images. Further loading results in generation of complementary strain localization macrobands (SLMB)

(Fig.3d, e, f) with their orientations deviating from the vertical one. Only at this stage of

the loading the optical images allow observing a primary crack oriented along the primary

SLMB. The primary crack starts propagating from a system of small cracks formed near a

central pore of the FOI, which is related to strain localization zone in Fig.3a.

The presence of inherent cracks offers easy routes for developing the primary crack

so that it can easily grow by breaking the only partial bonds between the crack-free segments. On the other side, these segments can independently slide against one another under loading and generate fragments. Such a process is analogous to that occurring during

external friction between two hard brittle bodies.”

*) These lines are not totally clear. Perhaps the authors should improve the description of the procedure also from a computational point of view.

*) Figure 3 and caption figure 3. The caption should be improved since it is definitely not clear. The authors should provide a self explanatory caption for interested readers. In optical images what are the dark areas ?

Lines: “The strain visualization maps obtained from specimens subjected to more intense
loading demonstrate more homogeneous strain distribution as compared to those discussed above (Fig.4). Both primary and complementary SLMBs still can be observed only
in Fig.4a and Fig.4b in contrast to quasi-homogeneous strain distributions in Fig.4c and
Fig.4d. The optical images show neither notable opening of the primary cracks nor forming extra ones. As a matter of fact, the image in Fig.4d allows suggesting some primary
crack closure.”

*) This paragraph is not clear, please explain better.

*) Figure 4 and its caption. See the previous comment for figure 3. The caption should be improved and self explainatory.

Paragraph “3.2.2. Estimation of microfracture retardation mechanism by rate-of-strain tensor (ROST) components”

Figure 6. Compression loading diagram and coordinate system

*) This paragraph and the figure should be moved to the “Methods” section, they are not results.

Figure 8. Distribution of  yy values (a) as computed along the middle line (Fig.7) and mean block size evolution with

the strain (b).

*) See the previous comments for figure 3 and 4. Please improve the caption in order to make it self explanatory.

Paragraph : “3.2.2. Fracture-to-compaction banding transition”

Lines: “Given that the displacement field is homogeneous the strain  values in each point of

the FOI must be the same value too. Strain tensor components and vector field components will be functions of their coordinates as follows:” , etc

*) Please, insert this part within the “Methods” section

Figure 10 and its caption

*) See the previous comments with respect to the other figures. In addition, perhaps the size of the figure should be adjusted.

*) The Discussion section is lacking.

References

*) The style of the references should be made homogeneous.

Author Response

Please, see the attached file "Response to Reviewer 1"

Reviewer 2 Report

This is an interesting work that expands our understanding of the deformation of porous ceramic materials. The researchers used modern experimental research methods as well as mathematical modeling. 
However, I think the work would be more interesting for readers if the authors add directly to the conclusions a brief indication of the applied application of the discovered results.
I also consider it necessary to point out that the article is not framed in accordance with the requirements of the journal. In particular, it was necessary to number the lines.

Author Response

This is an interesting work that expands our understanding of the deformation of porous ceramic materials. The researchers used modern experimental research methods as well as mathematical modeling. 
However, I think the work would be more interesting for readers if the authors add directly to the conclusions a brief indication of the applied application of the discovered results.
I also consider it necessary to point out that the article is not framed in accordance with the requirements of the journal. In particular, it was necessary to number the lines.

A: The structure of the manuscript has been improved.

Reviewer 3 Report

the manuscript should be impoved to lead a better understanding:

  • in abstract part, it is necessary to precise the loading mode: unidirection compression; the second sentence "Alumina ceramics contained 50 vol.% of 110 μm and 3 μm pores as well as discontinuities segmenting the sample into 220 μm blocks or segments" is not comprehensive;
  • in introduction part, it is necessary to indicate explicitly the originality and the specificity of this study; it is important to define cleanly the physical and mechanical meaning of "quasi plastic behavior"; 
  • in paragraph 2.1, it is important to precise the sample size, the loading surface/section of sample, the method for displacement recording. It is also necessary to discuss the influence of the rectangular plane comparing to sample mechanical loading state and the associated DIC errors;
  • in caption of figure 2, it is important to precise the "compression" loading, and the give the whole name of UCS
  • in figure 3 and 4, it is necessary to put a scale for each photography; and indicate the images have been obtained in-situ or ex-situ;
  • in caption of figure 5, it is necessary to precise the loading condition: in-situ or no, strain ratio etc...;
  • in caption of figure 7 and 9, it is necerssary to indicate the whole name of ROST;
  • in figure 8(a), it is necessary to precise what corresponding all curves;
  • in caption of figure 10, it is necerssary to indicate the whole name of MSD;
  • in caption of figure 8,9 and 10, it is important to precise the mechanical state, in-situ or ex-situ.
  •  
  •  

Author Response

Please, see the attached file Response to Reviwer 3

Round 2

Reviewer 1 Report

Title: “In-situ investigation of strain localization in sintered porous segmented aluminа”

General comment: This work has been reviewed according to the suggestions of this reviewer. Nevertheless, formally, several points should be still reworked in order to improve the quality of the work. The current version of the manuscript is like a “patchwork” and should be made homogeneous. In particular, text fonts and formulas for Equations should be inserted in a correct way within the text, while the figure captions and the figures size should be proportional to the main text. Please rework the manuscript accordingly.

Author Response

Thank you. The manuscript has been revised in accordance to the reviewer's comments

Reviewer 3 Report

in the revised version of the manuscirpt, author has made necessary corrections including most of remarks from reviewers.

Just one thing should be precised: "DIC" should be defined correctly in abstract and in introduction part.

Author Response

Thank you. The abbreviation has been disclosed in accordance to the  comment